# An Engineered Hybrid Protein from *Dermatophagoides pteronyssinus* Allergens Shows Hypoallergenicity

**DOI:** 10.3390/ijms20123025

**Published:** 2019-06-21

**Authors:** Dalgys Martínez, Marlon Munera, Jose Fernando Cantillo, Judith Wortmann, Josefina Zakzuk, Walter Keller, Luis Caraballo, Leonardo Puerta

**Affiliations:** 1Institute for Immunological Research, University of Cartagena, Cartagena 130000, Colombia; dmartinezdelaossa@unicartagena.edu.co (D.M.); marmunera@unicartagena.edu.co (M.M.); josefcantillo@gmail.com (J.F.C.); jzakzuks@unicartagena.edu.co (J.Z.); lcaraballog@unicartagena.edu.co (L.C.); 2Division of Structural Biology, Institute of Molecular Biosciences, BioTechMed, University of Graz, 8036 Graz, Austria; judith.wortmann@uni-graz.at (J.W.); walter.keller@uni-graz.at (W.K.)

**Keywords:** recombinant allergen, hybrid protein, allergy, IgE, allergen immunotherapy

## Abstract

The house dust mite (HDM) *Dermatophagoides pteronyssinus* is an important risk factor for asthma and rhinitis. Allergen specific immunotherapy that is based on recombinant proteins has been proposed for the safer and more efficient treatment of allergic diseases. The aim of this study was to design and obtain a hybrid protein (DPx4) containing antigenic regions of allergens Der p 1, Der p 2, Der p 7, and Der p 10 from this mite. DPx4 was produced in *Escherichia coli* and its folding was determined by circular dichroism. Non-denaturing dot-blot, ELISA, basophil activation test, dot blot with monoclonal antibodies, ELISA inhibition, and cysteine protease activity assays were performed. Mice that were immunized with DPx4 were also analyzed. We found that DPx4 had no cysteine protease activity and it showed significantly lower IgE reactivity than Der p 1, Der p 2, and *D. pteronyssinus* extract. DPx4 induced lower basophil activation than Der p 2 and the allergen extract. Immunized mice produced IgG antibodies that inhibited the binding of allergic patient’s IgE to the allergen extract and induced comparatively higher levels of IL-10 than the extract in peripheral blood mononuclear cells (PBMC) culture. These results suggest that DPx4 has immunological properties that are useful for the development of a mite allergy vaccine.

## 1. Introduction

IgE antibodies to normally innocuous environmental antigens, such as house dust mite (HDM), pollens, and animal dander, mediate respiratory allergic diseases. HDMs are an important source of inhalant allergens and inducers of allergic respiratory symptoms worldwide, particularly in tropical and sub-tropical regions, where they have the most favorable climatic conditions for their growth. The mite species *Dermatophagoides pteronyssinus* is very common in these regions [1,2]. Several allergens from this species have been characterized, showing different capabilities of sensitization in atopic individuals [1,3]. So far, allergen specific immunotherapy (SIT) with whole allergen extract is the only disease that modifies treatment of allergy [4,5]. However, this approach has some disadvantages, such as great variation in composition, missing of important allergens, and the inclusion of non-relevant molecules [6].

Recombinant allergens or their modifications exhibiting reduced allergenic activity have been proposed for more effective SIT [7]. Thus, a variety of recombinant products have been designed, ranging from peptides with T cell epitopes, mosaic, mutants, constructs of non-allergenic peptides fused to a carrier protein, and hybrid allergen molecules [8]. In animal models and human preclinical studies, these constructs may induce tolerance to natural allergen exposure [9,10]. The concept of hybrid molecules that are based on allergen-derived fragments has been applied for the construction of hypoallergenic proteins to treat allergy that is induced by complex allergen sources, such as grass pollen [11,12], bee venom [13], and HDM [14,15,16].

Der p 1 and Der p 2 are major allergens [17,18,19], therefore they are appropriate for designing reagents for diagnosis and SIT. Der p 7 and Der p 10 are allergens with a lower frequency of sensitization, but they are important inducers of allergy in some regions [20,21]. In this study, we report the design, production, purification, and immunologic characterization of a hybrid protein that is made up by segments of the four above-mentioned allergens that could be useful for the development of a HDM vaccine.

## 2. Results

### 2.1. DPx4 was Obtained as a Partly Folded Protein

The DPx4 protein was recovered from the inclusion bodies, after treatment with 8 M urea. It was successfully solubilized by successive dialysis against L-Cysteine-Cystine buffer with L-Arginine. The protein migration in SDS-PAGE was consistent with the theoretical molecular mass of 27.2 kDa (Figure 1A). The circular dichroism spectroscopy (CD) spectrum of the hybrid molecule showed a partially folded protein with a minimum at 215 nm, which was indicative of a high beta strand content (Figure 1B). The dynamic light scattering (DLS) analysis revealed that DPx4 is highly aggregated in solution in an oxidized form, as well as under reducing condition (Appendix A). From DLS data it can be estimated that aggregates consist of approximately 700 to 2000 molecules of DPx4 in the reduced and the oxidized state, respectively. In contrast, the natively folded Der p 1 has been shown to exist as a monomer in solution [22]. The protein that was stored under physiological conditions and analyzed with SDS-PAGE showed no visual degradation over a period of three weeks (Figure 1C). In contrast, the CD spectrum of nDer p 1 shows two minima at 208 nm and 220 nm, which were indicative of significant α-helical content. This is also confirmed by the mixed α/β-fold that was found in the crystal structure of Der p 1 (25% α-helix, 21% β-sheet; Appendix A).

The primary structure of DPx4 was modeled to a three-dimensional (3D) structure using the three-dimensional structure of Der p 1 (PDB 3F5V) as the template. The global and local structure prediction showed good quality, as indicated by a “Z” value of −0.78 according to the ProSA-web program [23]. Figure 2A indicates the order of segments in the DPx4 primary structure. Figure 2B shows the position of each segment in the cartoon representation of the model, and Figure 2C illustrates the position of each segment on the surface model.

### 2.2. DPx4 has Lower IgE Reactivity than Major Allergens of D. Pteronyssinus and the Allergen Extract

The frequency of IgE reactivity to DPx4 among the *D. pteronyssinus* allergic patients was 45% (41/90), whereas, in non-allergic individuals, it was 18% (10/55). In non-denaturing dot-blot assays, twenty-six sera from the battery of allergic sera showed strong IgE reactivity to *D. pteronyssinus* extract. In contrast, only three sera (#19, #20, and #26) were reactive to DPx4. Most of these sera showed IgE reactivity to Der p 2 and some of them also to Der p 1. Sera from three non-allergic individuals and the buffer alone showed no IgE reactivity to any of the dotted allergens (Figure 3A). The small number of sera with IgE reactivity to Der p 1 is unexpected, because a higher frequency of reactivity to this allergen in mite allergic patients from this region has been reported [18]. However, the low number of sera used in the dot blot assay (*n* = 26) could be one of the reasons for this result.

Specific IgE levels to DPx4 were significantly lower than those to the *D. pteronyssinus* extract in 90 serum samples from allergic patients, *p* < 0.001, (Figure 3B). In 32 sera from random-selected allergic patients, the IgE levels to DPx4 were also significantly lower than those to Der p 1 and Der p 2, *p* < 0.001 (Figure 3C). By ELISA, the IgG reactivity to DPx4 was detected in all sera from allergic and non-allergic individuals, resulting in higher antibody levels in the allergic group (0.944 vs. 0.826, *p* < 0.004) (Appendix A).

### 2.3. DPx4 Shares Antigenic Epitopes with Allergens from D. pteronyssinus

The ELISA inhibition assay showed that, at 25 μg/mL, DPx4 inhibited the IgE reactivity of serum pool A to Der p 1 and Der p 2 by 38% and 26%, respectively (Figure 4A,B). *D. pteronyssinus* extract inhibited the IgE reactivity to DPx4 at the highest concentration by 78.3%, which was similar to the rate that was obtained in the homologous inhibition (Figure 4C), while DPx4 inhibited IgE reactivity to *D. pteronyssinus* extract up to 35.1% (Figure 4D). These results indicate that an important number of the IgE-binding epitopes of allergens that are present in the extract and purified allergens are on DPx4. No inhibition was observed while using BSA as inhibitor. In addition, these results are supported by data from the dot blot assay using the monoclonal antibodies anti-Der p 1 and anti-Der p 2, which showed the reactivity of these antibodies against DPx4 (Appendix A).

### 2.4. DPx4 Lacks Cysteine Protease Activity 

The cysteine protease assay showed the following value of specific enzymatic activity: 67.040 × 10^−4^ µMol/min for papain, 48.691 µMol/min for DPx4, and 76.5 µMol/min for FABP4, which indicated that DPx4 has around 7000 fold less activity as compared to papain used as positive control. The enzymatic activity of DPx4 was similar to that obtained with the recombinant FABP4, a protein lacking protease activity.

### 2.5. DPx4 Induced Lower Basophil Activation than Der p 2 and D. pteronyssinus Extract

The incubation of basophils from allergic patients with 10 µg/mL of Der p 2 or *D. pteronyssinus* extract induced significant upregulation of CD203c expression (SI from 1.8 to 10), whereas no upregulation was obtained with the same concentration of DPx4 (SI 1.0 to 1.4) (Figure 5). No activation was observed in the cells from a non-allergic subject. Anti-human IgE antibodies induced basophil activation in all samples from patients. No activation was obtained with PBS. These results suggest that DPx4 has lower allergenic activity than Der p 2 and, as expected, *D. pteronyssinus* extract. 

### 2.6. DPx4 Induced Higher Levels of IL-10 and Lower Levels of IL-5 than Mite Extract in Allergic Patient´s PBMCs

The stimulation of peripheral blood mononuclear cells (PBMC) cultures from HDM allergic patients with DPx4 induced significant higher levels of IL-10 and lower levels of IL-5 as compared to that observed with *D. pteronyssinus* (Figure 6A,B). DPx4 and *D. pteronyssinus* extract did not induce IFN-gamma production, observing similar levels to the non-stimulated condition (Figure 6C). IL-4 levels were, in all cases, below the limit of detection.

### 2.7. DPx4 Induced IgG Antibodies that Inhibited Patient’s IgE Reactivity to D. pteronyssinus Extract

Mice that were immunized with DPx4 produced higher levels of IgG2a than IgG1 with a ratio IgG1/IgG2a of 0.7 (Figure 7A), which suggested that DPx4 promoted a mixed Th1/Th2 immune response with predominance of IgG2a antibodies. It is important to highlight that the IgG antibodies reacted to the allergen extract (Figure 7B).

When testing whether the DPx4-specific IgG antibody could inhibit the human IgE binding *to D. pteronyssinus* extract, it was found that the IgG antibodies produced high percentage of inhibition, ranging from 39% to 78%, which indicated that DPx4 can induce the production of blocking antibodies. No inhibition was observed when testing serum from a non-allergic individual (Table 1).

## 3. Discussion

The rationale behind the design of the hybrid protein was to obtain one molecule containing antigenic regions from several *D. pteronyssinus* allergens with the inclusion of at least the major allergens Der p 1 and Der p 2, plus other allergens that also contribute to the sensitization in allergic individuals. This approach was based on the information of HDM allergens that are available at the time of designing the molecule (2010). Group 1 and 2 are clinically relevant HDM allergens and they are considered to be essential components for HDM allergy vaccines [17,20,24]. Several promising hypoallergenic molecules derived from group 1 and 2 mite allergens have been developed, showing lower capacity than wild type allergen to bind patient´s IgE and reduced allergenic activity [14,15,16]. The design of the hybrid protein was also guided by reports that combinations of Der p 1, Der p 2, and a few additional allergens, including group 10 mite allergen, increased the probability of appropriate diagnosis [18,25]. 

Der p 7 generally exhibits low levels and frequency of IgE binding. However, it has been reported that it can interact with LPS activated dendritic cells, and that it might stimulate the innate immune response [26]. Although this allergen shows lower frequency of sensitization than Der p 1, Der p 2, and Der p 10, it is an important inducer of allergic reaction in some patients. Under this premise, we hypothesized that an important number of mite allergic patients could be diagnosed with a combination of these particular allergens and that a single molecule representing their epitopes, might be a promising candidate for the development of a HDM allergy vaccine. In addition, during the process of design, we noticed that, when the short segments of Der p 7 present in DPx4 were introduced, there was an improvement of the structural parameters of the 3D structure model, such as the increase in the number of amino acids in the favored regions, according to the Ramachandran plot, and better Z-score value for global and local quality prediction of the structure according to ProSA-web [23].

The lower IgE reactivity of DPx4, as compared with the *D. pteronyssinus* extract, may be the consequence of the lower number of IgE epitopes, the non-appropriate geometric arrangement of them affecting the affinity to IgE antibodies, and/or non-optimum cross-linking to IgE receptors on the effector cells. These possibilities could also explain why the specific IgE levels to DPx4 were also significantly lower than those to Der p 1 and Der p 2. However, since the dot blots with monoclonal anti Der p 1 and anti Der p 2 antibodies, and inhibition assays with human serum pool, indicated the presence of specific B-cell epitopes in DPx4, the position and orientation of the epitopes in the new structure could modify their avidity, affecting the strength of reactivity. Consequently, DPx4, but not Der p 2, failed to induce basophil activation in allergic patients. The low percentage of IgE inhibition to *D. pteronyssinus* extract and Der p 2 by DPx4, and the results of ELISA and the basophil activation test indicate a lower allergenic activity when compared to mite extract and purified major allergens, suggesting a reduced risk of the hybrid protein to induce IgE mediated side effects. Therefore, when regarding the potential development of a HDM vaccine, DPx4 could provide a safer treatment when compared to the current approach of allergen specific immunotherapy based on the whole mite extract.

A mechanism that seems to contribute to the low allergenic activity of recombinant proteins that are derived from native allergens is the degree of aggregation. For instance, the protein Bet v 1d [27], the Bet v 1 trimer [28], Phl p 5-Bet v 1 hybrid [29], and a hypoallergenic derivative of Fel d 1 [30] showed the formation of aggregates that seem to favor their hypoallergenic properties supporting a promising role for the development of allergy vaccine. Aggregation, as shown in the case of DPx4 (Appendix A), likely influences its allergenic properties and it may explain, in part, its comparatively lower allergenic activity and the IgE binding capacity. However, the formation of aggregates could affect the stability and reproducibility of the protein. In this regard, in the future it will be necessary to maintain good manufacturing practice and standardize the aggregation behavior and the immunogenic properties of the protein. 

The reactivity of the anti-Der p 1 monoclonal antibody against DPx4 suggests that the hybrid protein keeps at least one of the B-cell epitopes from the native allergen Der p 1, which involves the a.a residues Glu^13^, Arg^17^, Gln^18^, and Asp^198^ [31,32]. The presence of shared epitopes is also supported by the 35.1% of inhibition of IgE reactivity to *D. pteronyssinus* extract by DPx4 (Figure 4D). Anti Der p 2 monoclonal antibody (Dpx-A9) also reacts against DPx4 (Appendix A), indicating the presence of specific epitopes. The residues 47 to 52 of Der p 2, which belong to an epitope recognized by this monoclonal [33], were incorporated into DPx4. However, the inhibition of IgE reactivity to Der p 2 by DPx4 was 26%, as compared to the 38% of inhibition that was obtained with Der p 1 (Figure 4A,B). This difference can be explained by a lower percentage of Der p 2 in the structure of DPx4.

The cysteine protease activity of Der p 1 is involved in the pathogenesis of mite allergy by inducing a number of changes that affect innate immunity and epithelial inflammation [34]. Recently, an allergic airway inflammatory effect has been associated with cysteine protease activity of *D. pteronyssinus* by the modulation of IL-33 [35]. In mice that were immunized with Der p 1 inactivated by the cysteine protease inhibitor E-64, the production of total and specific IgE to Der p 1 was significantly reduced, which suggested that the immune response to this allergen can be modulated by altering its proteolytic activity [22,36]. In fact, the cysteine protease activity of Der p 1 has been assayed for designing a novel allergen vaccine [37]. Interestingly, we found that DPx4 lacks protease activity in spite of having 79% of identity in the amino acid sequence with Der p 1 and containing the sequence motif for protease activity. This finding could be due to DPx4 adopting a fold that prevents the uptake or processing of the substrate. This feature might represent an advantage for SIT because it would allow for the use of higher doses, but still avoid the risk of adverse effects.

The IgG antibodies that are produced by the immunized mice reacted against the allergen extract, which is important, because the allergen-specific IgG antibodies that are produced in the course of SIT are related to the efficacy of treatment. This effect may result from several mechanisms, such as the inhibition of basophil degranulation [38] and the IgE facilitated allergen presentation, as well as the consecutive T-cell activation [39]. In addition, the blocking properties of these antibodies have been demonstrated to mediate their anti-allergic effect [40,41]. The immunization of BALB/c mice with DPx4 induced IgG antibodies that inhibited up to 78% of patient IgE-binding to the mite extract. This high inhibitory effect might be because DPx4 induces IgG antibodies with different specificities, which represent a wide repertoire of epitopes from the allergen extract. There is no natural exposure to DPx4, since it is a synthetic molecule, therefore the IgG reactivity with sera from HDM allergic patients suggests that, in case of being used for immunotherapy, DPx4 could induce, in the context of a T-cell collaboration, an immune response with important participation of IgG antibodies, which have been associated with the beneficial effects of allergy vaccines [42].

Similar to other hybrid molecule of 22.8 kDa that was developed by our group (MAVAC-BD-2) [43], DPx4 induced IgG blocking antibodies in immunized mice, showed comparatively lower human serum IgE reactivity and basophil activation than the purified allergens and mite extract. However, there are major differences between these molecules in allergen representation and antigenic composition; DPx4 contains antigenic segments of four allergens from only *D. pteronyssinus*, with a larger contribution from Der p 1, while MAVAC-BD-2 contains antigenic segments of four allergens from this mite species and three allergens from *B. tropicalis*, with the largest contribution of Der p 2 and Blo t 5. In regards to exposure to allergens from *B. tropicalis* being mainly limited to tropical and subtropical regions, we can speculate that MAVAC-BD-2 could be more effective for allergic patients from these regions showing sensitization to both mite species. On the other hand, DPx4 effective on worldwide allergic patients that are sensitized to *D. pteronyssinus* allergens.

In PBMC cultures from HDM-allergic patients, stimulation with DPx4 induced higher levels of IL-10 than the allergen extract, and lower IL- 5 levels accompanied it. These results allow for speculating that DPx4 induces an immune response different to the classical allergic Th2 response, which predicts the efficacy for allergy treatment [44]. However, some limitations of these experiments should be mentioned; first, the low number of individuals and cytokines analyzed; second, we could not to define the source of IL-10, because this cytokine is produced by different cells that are present in PBMC [45]. Therefore, the type of IL-10 producing cell stimulated by DPx4 remains to be established.

Several aspects indicate that DPx4 could be useful for the development of a mite allergy vaccine; a) being a defined structure having allergen specificity, it can be applied in a tailored immunotherapy, b) the reduced IgE binding and basophil activation, as well as the ability to induce protective IgG antibodies, are desirable properties of hypoallergenic molecules for SIT, c) it contains segments of two of the most sensitizing HDM allergens, Der p 1 and Der p 2, as well as segments of Der p 7 and Der p 10, which expands the range of patients that can benefit from SIT, d) it lacks protease activity, which is a key factor for inducing allergic inflammation [35]. The above-mentioned properties suggest that DPx4 could be useful for SIT for HDM allergy. Furthermore, the strategy of in silico design of single molecules containing epitopes from different allergens could be more generally applied for developing allergy vaccines.

Recently, after the design and obtaining of the DPx4 molecule, Der p 23 was described as an allergen inducing high frequencies of sensitization in allergic people from Europe (Austria, France, and Germany) [46] and United States of America (USA) [47] and suggested as allergen to be considered for immunotherapy of mite allergy. It seems reasonable that the IgE reactivity that was displayed by this allergen in the referred populations merits including Der p 23 in the design of novel hybrid molecules. 

## 4. Materials and Methods

### 4.1. Human Sera Used in This Study

Sera from 90 HDM allergic patients and 55 non-allergic healthy volunteers were selected from a well-characterized dataset of patients that were recruited from the Social Security Clinic and public health centers in Cartagena, Colombia [48]. The patients that were in this study were asthmatic and sensitized to *D. pteronyssinus,* as determined by a positive skin prick test (SPT) and serum positive IgE levels to allergen extract by ELISA. Asthma was defined based on the Global Initiative for Asthma (GINA) criteria and was confirmed by a physician that belonged to the research staff, sustained on a clear clinical history. All of the controls were subjects without allergy history with negative results to SPT and mite-specific IgE tests. For SPT a *D. pteronyssinus* extract that was obtained from Greer Laboratories (Lenoir, NC, USA), at 10,000 AU/mL was used. For the basophil activation test, heparinized blood samples were obtained from seven allergic patients and one non-allergic control. The Bioethics Committee of University of Cartagena, Colombia approved this study (approval date 15 March 2009, No. 16-04-2019). Project code: 110748925408. Written informed consent was obtained from all patients and the non-allergic controls. 

### 4.2. HDM Extract and Purified Allergens

A whole mite culture of *D. pteronyssinus*, which Dr. Enrique Fernandez-Caldas kindly donated, was used to prepare the allergen extract, as previously described [49]. Whole mite culture was defatted for 4 h with anhydrous ether in a Soxhlet extractor and then dried at room temperature (RT). Defatted mites were extracted (1: 20 wt/vol) in 0.1 mol/L of ammonium bicarbonate for 24 h at 4 °C. The extract was clarified by centrifugation at 18,000 r.p.m. and dialyzed overnight at 4 °C against deionized water with Spectra Por 3 membranes with 3500 molecular weight cutoff (Spectra/Dialysis membranes, Houston, TX, USA). The dialyzed extract was centrifuged, filtered with 0.22 µm membranes, and lyophilized and stored at 4 °C until use. 

Codon-optimized gene sequence of Der p 2 (Uniprot P49278) and nucleotide sequence of FABP4 (UniProt P15090) were synthetized and then subcloned into pET45b+ vector by GenScript (Piscataway, USA). Der p 2 was expressed in Origami™ B (DE3) competent cells Novagen (Wisconsin, USA), and FABP4 was expressed in *Escherichia* coli (*E. coli*) BL21 (DE3) competent cells (Invitrogen Corporation, Carlsbad, CA, USA). The cells were transformed by electroporation and selected on Luria Bertani (LB) agar plates containing ampicillin. The transformed cells were grown at 37 °C to an OD600 of 0.5–0.8. Subsequently, the culture was induced with 1 mM Isopropyl-1-β-thiogalactopyranoside (IPTG) by incubation for 5 h and 4 h at 37 °C, respectively. Cells were harvested by centrifugation at 5000 r.p.m. during 30 min. at 4 °C. For Der p 2, the pellet was re-suspended in native buffer (50 mM NaH_2_PO_4_, 300 mM NaCl), and sonicated on ice while using an ultrasonic homogenizer. Insoluble debris was removed by centrifugation and the supernatant was loaded to a Ni-NTA resin (Invitrogen, Carlsbad, CA, USA) for 1 h, washed with native buffer plus 20 mM imidazole, and eluted with native buffer plus 250 mM imidazole. Recombinant protein was dialyzed against water and lyophilized and stored at −20 °C. For FABP4, the pellet was re-suspended in denaturing buffer (8 M urea, 100 mM NaH_2_PO_4_, and 10 mM Tris, pH 8.0) and sonicated, as above. Insoluble debris was removed by centrifugation and the supernatant was loaded to a Ni-NTA affinity column under hybrid conditions following the instructions from the manufacturer (Invitrogen, Carlsbad, CA, USA). Recombinant protein was dialyzed against a buffer (50 mM NaH_2_PO_4_ and 0.5 M NaCl, pH 8.0).

Natural Der p 1 (NA-DP1) and natural Bet v 1 (NA-BV1-1) were obtained from Indoor Biotechnologies (Charlottesville, VA, USA).

### 4.3. Construction of the Hybrid Protein DPx4 

Four segments of Der p 1.01 (GenBank ABV66255.1), two segments of Der p 2.01 (GenBank ABY53034.1), two segments of Der p 7.01 (GenBank P49273.1), and one of Der p 10.01 (GenBank ABB52642.1), were selected to be placed in a single molecule. Segment selection was based on their antigenic value, as indicated by literature search [50,51,52], and as suggested by using prediction tools [53]. The segments were assembled, from the N-terminal to the C- terminal ends, in the following order: amino acids (a.a) 42–52 of Der p 2, a.a 157–165 of Der p 7, a.a 23–44 of Der p 1, a.a 93–106 of Der p 10, a.a 63–112 of Der p 1, a.a 103–118 of Der p 2, a.a 125–204 of Der p 1, a.a 179–187 of Der p 7, and a.a 215–223 of Der p 1. The full amino acid sequence of designed protein is provided in Appendix A. The molecule containing the selected segments was designed and the 3D structure modeled while using I-TASSER [54]. ProSA-web and Ramachandran plot were used to validate the 3D model structure [23]. The PyMOL Molecular graphic System was used to visualize structures and generate images [55]. The synthesis of the nucleotide sequence coding for the designed protein was ordered from GenScript, (Piscataway, NJ, USA) with codon optimization for the efficient expression in *E. coli*. The sequence was subcloned into the vector pET45b+ using the Kpn I and Pml I restriction sites, with the N-terminal 6xHis-tag next to the DPx4 sequence for affinity purification.

### 4.4. Expression and Molecular Characterization

The plasmid containing of DPx4 insert was transformed into competent *E. coli* BL21 (DE3) cells (Invitrogen Corporation, Carlsbad, CA, USA) by electroporation. The expression and purification was performed, as described elsewhere [43]. In brief, the culture was induced with 1 mM IPTG for 4 h at 37 °C. The protein that was obtained in inclusion bodies was solubilized by incubating in denaturing buffer (8 M urea, 100 mM NaH_2_PO_4_, and 10 mM Tris, pH 8.0) and purified while using a Ni-NTA affinity column. For refolding, the hybrid protein was dialyzed against renaturalization buffer (10 mM EDTA, 0.5 M L-Arginine, 5 mM L-Cysteine, 1 mM Cystine, 100 mM Tris pH 8.0). Endotoxin was removed by ToxinEraser^TM^ Endotoxin Removal Resin Kit (Piscateway, NJ, USA). 

The purified hybrid protein was analyzed by SDS-PAGE with 15% polyacrylamide separation gels using a Miniprotean II System (BioRad, USA) under reducing conditions, according to Laemmli, and stained with Coomassie Brilliant Blue R-250 (BioRad, Hercules, CA, USA). For stability studies, DPx4 was stored under physiological conditions at 4 °C and the samples were taken at different time points (0 days, 3 days, and 1, 2, and 3 weeks) for analysis with a 12% SDS-PAGE.

The aggregation state of DPx4 in 20mM Na_2_PO_4_, pH 8.0 (oxidized form), and in the same buffer, including 1mM DTT (reduced form), was investigated by dynamic light scattering (DLS) while using a Zetasizer nano ZS (Malvern, Worcestershire, UK). After centrifugation at 14,000× *g* for 30 min., 50 µL sample was transferred into a 3 mm quartz cuvette. The measurements were carried out at 20 °C with 173° backscatter with a size range of 0.6 to 6000 nm. Three subsequent measurements for each sample with 15 runs of 30 sec per run were performed. The molecular weight was estimated by a mass vs. size calibration curve.

The structural integrity of the DPx4 was evaluated while using circular dichroism spectroscopy. Measurements were carried out at concentration of 0.09 mg/mL in a 0.1 cm quartz cuvette while using a Jasco J-715 CD-spectrometer (Japan Spectroscopic Co., Tokyo, Japan). Far UV spectra were recorded in the range of 190 to 260 nm. The spectra were baseline, corrected by buffer subtraction and then converted to mean residue ellipticities (Θ) at the given wavelengths. The calculation of the secondary structure content was performed with Dichroweb [56]. 

### 4.5. Antibody Reactivity of DPx4 Determined by Non-Denaturing Dot-Blot and ELISA Assays

26 sera were randomly chosen from the allergic patients to compare the IgE reactivity to DPx4 with that of Der p 1, Der p 2, and *D. pteronyssinus* extract for non-denaturing dot blot experiments, as described previously [43]. Three sera from non-allergic individuals and buffer alone were used as the controls.

For the ELISA assay, Immulon-4 microtiter plate wells (Dynatech, Chantilly, VA, USA) were coated with 0.5 µg of DPx4, Der p 1, Der p 2 or 5 µg of allergen extract in sodium carbonate/bicarbonate buffer at 4 °C overnight (ON). After washing with phosphate buffered saline (PBS), 0.1% Tween 20 (PBS-T) wells were blocked with PBS-T, 1% bovine serum albumin (BSA) for 3 h at RT. After washing, 100 μL of human serum diluted 1:5 or 1:100 in PBS-T-BSA for IgE and IgG, respectively, were added and then incubated ON at RT. The bound IgE and IgG antibodies were detected by incubation with 100 μL of alkaline phosphatase-conjugate anti human IgE (Sigma A3525) or anti-IgG (Sigma A3187). The substrate p-nitrophenyl phosphate (Sigma) diluted in 10% diethanolamine was added for the development of reactivity, and OD was measured in an ELISA reader at 405 nm [43]. The quantification of IgE was performed by using the standard curve of human IgE (Merk Millipore, Billerica, MA, USA). IgE diluted in 0.2 M carbonate buffer at two-fold dilution (125 ng/mL to 0.1 ng/mL) was used for coating the wells.

### 4.6. B-Cell Epitopes Analysis 

To investigate whether relevant B cell epitopes of Der p 1, Der p 2, and allergen extract were present on DPx4, dot-blot assays with monoclonal antibodies and ELISA inhibition assays were performed. For dot-blot assays, aliquots containing 2 µg of DPx4, Der p 1, Der p 2, or BSA were dotted onto 0.45 μm nitrocellulose membrane. The dot-blots were blocked with PBS-T, 5% skimmed milk for 1 h at RT, and then incubated with either anti-Der p 1 monoclonal antibody (4C1, Indoor Biotechnologies) or anti-Der p 2 monoclonal antibody (Dpx-A9, Indoor Biotechnologies). Subsequently, the membranes were washed and bound mouse monoclonal IgG antibodies were detected by incubation with anti-mouse IgG (Sigma A1418) diluted 1:2.000 in PBS-T for 1 h at RT, and then developed, as described above.

Two serum pools were prepared for IgE ELISA inhibition assay (Table 2). For the inhibition of the IgE reactivity to Der p 1 and Der p 2 by DPx4, MaxiSorp plate wells (Nunc, Thermo Fisher) were coated with 0.5 µg of Der p 1 or Der p 2, and blocked, as described above. Pool A diluted 1:5 in PBS-T-BSA was incubated ON at 4 °C with two concentrations (0.1 and 25 μg/mL) of DPx4, Der p 1, Der p 2, or BSA. For the inhibition of IgE reactivity to *D. pteronyssinus* by DPx4 or viceversa, the wells were coated with *D. pteronyssinus* extract or DPx4 and blocked, as described in the ELISA section. Pool B diluted 1:5 was incubated ON at 4 °C with several concentrations (0.001, 0.01, 0.1, 1, and 5 µg/mL) of DPx4, *D. pteronyssinus*, or BSA. The bound IgE antibodies were detected with alkaline phosphatase-conjugate anti-human IgE according to the ELISA protocol described above.

### 4.7. Basophil Activation Test by Flow Cytometry

Peripheral blood was collected from seven selected HDM allergic patients with IgE reactivity to *D. pteronyssinus* and DPx4, and one non-allergic individual (Table 3). Basophil activation was measured by flow cytometry while using the Allergenicity Kit (Beckman Coulter, Inc. CA, USA) following the instructions from the manufacturer. A dose-response assay was performed with three of the seven samples using three concentrations (0.1, 1 and 10 µg/mL) of DPx4 and Der p 2 to define the more appropriate concentration of antigen for stimulation. The concentration of 10 μg/mL was chosen because it induced a clear effect on basophil with the antigen being used as reference (Der p 2). For analysis, 100 µL of blood was incubated with antigen. The basophils were gated based on the expression of CRTH2 marker and activation was assessed by the detection of CD203c. At least 500 basophils were counted in each assay. Allergen-induced upregulation of CD203c was expressed as the Stimulation Index (SI), (SI = MFstim/MFIcontrol). An SI value ≥ 2.0 was considered as a positive result.

### 4.8. Determination of Cysteine Protease Activity

We evaluated whether the hybrid protein conserved this activity, since DPx4 exhibits 79% of sequence identity with Der p 1, including the signature for cysteine protease activity. The protease activity was determined in a continuous rate (kinetic) assay while using the chromogenic substrate PFLNA (Sigma P3169), as described by Katsaros et al [57]. Fifty µg of DPx4 and 50 μg of papain (positive control) were incubated for 20 min. at 40 °C with activation buffer (0.1 mM cysteine, 0.1 M EDTA, 0.2 M PBS pH 7.0) in microtiter plate wells. Subsequently, substrate was added at concentrations of 0.1, 0.2, 0.3, 0.4, and 0.5 mM. The reaction rate was determined for 5 min. at 410 nm, while using a spectrophotometer (Spectra MAX 250, Molecular Device, Sunnyvale, CA, USA). The measurements were performed in duplicate.

### 4.9. Cytokine Production by PBMC in Response to DPx4

Previous to analysis purified DPx4 and mite extract were subjected to affinity chromatography while using the ToxinEraser™ Endotoxin Removal Resin Kit (Piscataway, NJ, USA). The concentration of LPS was determined using the ToxinSensor Chromogenic LAL Endotoxin Assay kit (Piscataway, NJ, USA). The content of LPS in the culture supernatant was 0.07 EU/mL for DPx4 and 0.02 EU/mL for mite extract. The PBMCs were obtained from fresh blood of six allergic patients by Ficoll (Sigma-Aldrich) density gradient centrifugation. The cells were cultured in RPMI with 10% heat-inactivated fetal bovine serum, 31 µg/mL penicillin, 50 μg/mL streptomycin, and 2 mM l-glutamine in 24 well plate (flat bottom plate; Costar, Cambridge, U.K.) at 1 × 10^6^ /mL per well in 500 µL of RPMI. The cell cultures were stimulated with DPx4 at final concentration of 1 µM or 25 µg/mL of *D. pteronyssinus* extract. Phytohaemagglutinin A (PHA) [5 μg/mL] was used as a positive control of immune cell activation. The cells were incubated for six days at 37 °C in a 5% CO_2_ humidified air atmosphere. The supernatants were collected after centrifugation and then stored at −70 °C until assessment of cytokine levels.

The levels of IL-5, IL-4, IL-10, and IFNγ were determined by flow cytometry in a FACSaria III while using human Th1/Th2 of Bio-Plex^TM^ cytokine assay (BD, CA, USA), according to the manufacturer’s protocol. 

### 4.10. Immunization of Mice with DPx4 and Analysis of Blocking IgE Reactivity by Their Sera

Six to eight-weeks old female BALB/c mice were immunized with 10 μg of DPx4 or PBS (negative control) being adsorbed to aluminum hydroxide adjuvant on days 0, 7, 14, and 21. One week after the last immunization, the mice were anesthetized with ketamine/xylazine and intranasally challenged with 10 µg DPx4 for three consecutive days. On day 33, the mice were sacrificed, and blood was drawn by cardiac puncture. Preimmune sera were obtained from the tail vein before the first intraperitoneal application and stored at −20 °C until use. The Ethical Committee of the University of Cartagena approved the experiments (approval date 06 February 2012, No. 42-2012). Project code: 110756934352.

DPx4-specific IgG1 and IgG2a in serum were determined by ELISA while using MaxiSorp plate wells (Nunc, Thermo Fisher) that were coated with 1 µg/mL of DPx4. In addition, we evaluated whether IgG antibodies induced by immunization reacted with mite extract by dot blot assay, using nitrocellulose membrane that was dotted with 5 µg of allergen extract, as described in [43]. An ELISA inhibition test was performed using MaxiSorp plate wells that were coated with 5 µg of *D. pteronyssinus* extract and serum pool from the immunized mice with DPx4 to test whether the IgG antibodies could inhibit the human IgE binding to *D. pteronyssinus* extract, as described previously [43].

### 4.11. Statistical Analysis

Analyses were performed using Graph Pad Prism software (San Diego, CA. USA). Wilcoxon rank test was used for two-group comparisons of paired continuous variables. Friedman’s test was used for multiple repeated measure comparisons. The data were expressed as mean ± SD, *p*-values less than 0.05 were considered to be statistically significant.

## 5. Conclusions

Our results suggest that DPx4 could be useful for development of HDM allergy vaccine. The hybrid protein exhibits low allergenic activity, induced mouse IgG antibodies that inhibited the binding of allergic patient’s IgE to the allergen extract and induced comparatively higher levels of IL-10 than the *D. pteronysssinus* extract in PBMC culture.

## Figures and Tables

**Figure 1 ijms-20-03025-f001:**
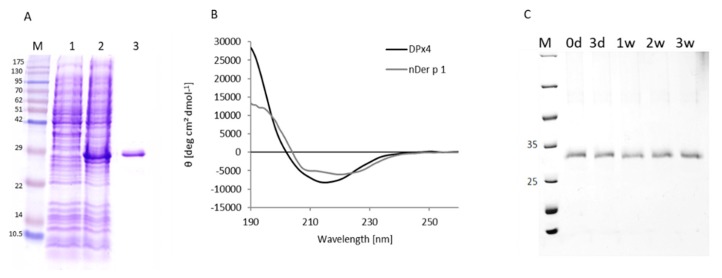
Structural analysis. (**A**). The gel stained with Coomassie brilliant blue shows a single band of 27 kDa corresponding to purified DPx4. M: Molecular weight marker. Lane 1: non-induced culture. Lane 2: culture induced with Isopropyl-1-β-thiogalactopyranoside (IPTG). Lane 3: Purified DPx4 protein. (**B**) Circular dichroism spectra of DPx4 and nDer p 1; spectra were recorded from 190 to 260 nm (x-axis) and circular dichroism (CD) values converted to mean-residue ellipticities (y-axis). (**C**) Stability of DPx4 upon extended storage (0–21 days).

**Figure 2 ijms-20-03025-f002:**
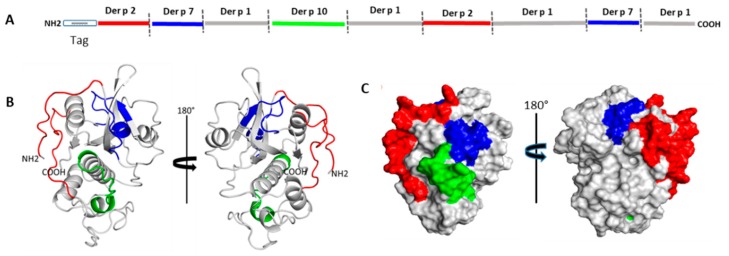
Schematic representation of DPx4 structure. (**A**). Order of segments in the DPx4 primary structure. (**B**) Cartoon representation, each segment is colored in the three-dimensional (3D) model in accordance to the antigenic segment defined above. (**C**) Surface color representation of the allergen segments included in the structure.

**Figure 3 ijms-20-03025-f003:**
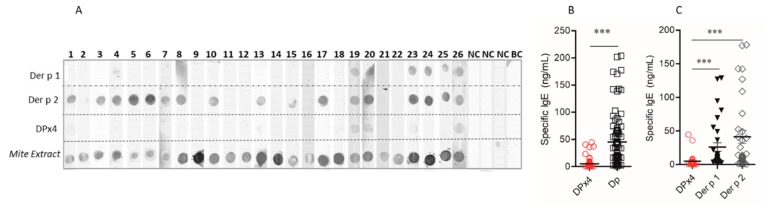
Serum IgE levels as detected by ELISA. (**A**) Non-denaturing dot blot showing the IgE reactivity of sera from 26 allergic patients (lanes 1–26), three non-allergic individuals (NC) and buffer alone (BC) in the nitrocellulose blotted proteins. (**B**) IgE reactivity to DPx4 and mite extract in 90 serum samples from allergic patients. (**C**) IgE reactivity to DPx4, Der p 1, and Der p 2 in sera from 32 random-selected allergic patients (*** *p* < 0.001).

**Figure 4 ijms-20-03025-f004:**
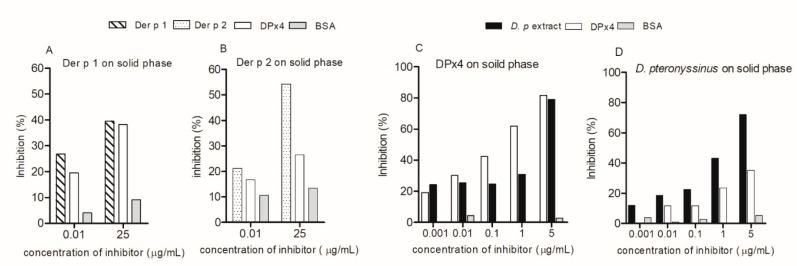
ELISA inhibition results. The percentage of inhibition of IgE reactivity of serum pools by DPx4 to Der p 1 (**A**), Der p 2 (**B**), and DPx4 (**C**). Inhibition of IgE reactivity to *D. pteronyssinus* extract by DPx4 (**D**). Homologous inhibitor and bovine serum albumin (BSA) were used as controls. Dp: *D. pteronyssinus*.

**Figure 5 ijms-20-03025-f005:**
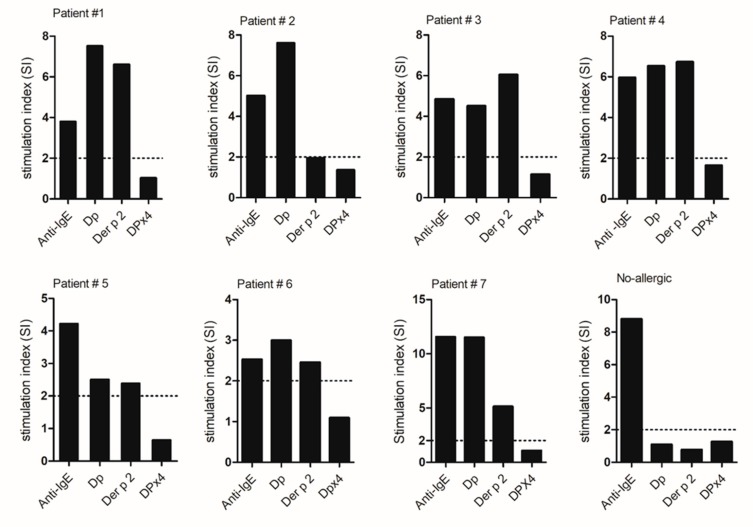
Basophil activation test. Stimulation index obtained in samples from seven allergic patients stimulated with anti-IgE, Dp (extract), Der p 2, and DPx4.

**Figure 6 ijms-20-03025-f006:**
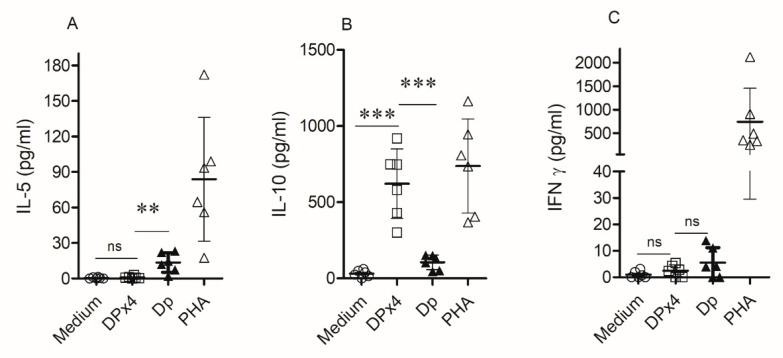
Cytokine response induced by DPx4 in cultures of peripheral blood mononuclear cells (PBMC) from six house dust mite (HDM) allergic patients. Culture supernatants were collected and levels of IL-5 (**A**), IL-10 (**B**), and IFNγ (**C**) were measured by flow cytometry. Results are expressed as the mean and their SD from the six samples. (** *p* < 0.05, *** *p* < 0.001).

**Figure 7 ijms-20-03025-f007:**
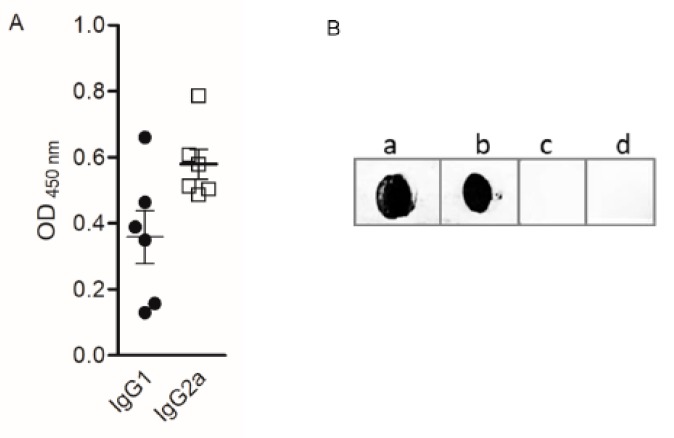
Reactivity of IgG antibodies induced by immunization of mice with DPx4. (**A**) Allergen-specific IgG1 and IgG2a levels by ELISA. (**B**) Sera from mice immunized with DPx4 were tested for IgG reactivity to dot-blotted *D.*
*pteronyssinus* extract (a), DPx4 (b), and Bet v 1 (c). No reactivity was observed when blotted hybrid protein was incubated with pooled mouse pre- immune sera (d).

**Table 1 ijms-20-03025-t001:** Inhibition of patient’s IgE binding to *D. pteronyssinus* by mouse IgG antibodies induced by immunization with DPx4.

	*D. pteronyssinus* on Solid Phase		
	OD Value		
Patients	Pre-Immune	Immunized	% Inhibition
**1**	2.564	1.232	52
**2**	2.963	1.095	63
**3**	2.658	0.939	65
**4**	2.421	0.963	60
**5**	0.695	0.258	63
**6**	0.486	0.185	62
**7**	0.315	0.193	39
**8**	2.014	0.901	53
**9**	0.683	0.147	78
**Mean**	2.524	1.164	59
**No-Allergic**	0.087	0.093	6

OD: Optical density.

**Table 2 ijms-20-03025-t002:** Characteristics of sera used for preparation of serum pools.

				IgE Levels (OD)	
	Code	Gender	DPx4	Der p 1	Der p 2
**Serum Pool A**	AC043	F	0.41	0.60	0.27
	AF33	M	0.33	0.43	0.29
			**DPx4**	***D. pteronyssinus***	
**Serum Pool B**	A676	F	0.77	0.62	
	A635	F	0.30	0.75	
	AC043	F	0.41	0.31	

OD: Optical density.

**Table 3 ijms-20-03025-t003:** Immunological characteristics of sera used in basophil activation.

		IgE Levels (OD)			Skin Prick Test > 3 mm	
Patients	Sex	*D. Pteronyssinus*	Der p 2	DPx4	Bt	Dp
1	M	0.83	2.38	0.21	+	+
2	M	0.79	0.55	0.50	+	+
3	F	0.58	0.82	0.15	+	+
4	M	2.34	0.63	0.20	+	+
5	M	0.70	0.15	0.13	+	+
6	F	0.32	0.30	0.29	+	+
7	M	0.58	0.14	0.25	+	+
No-Allergic	M	0.09	0.10	0.10	-	-

Dp: *D. pteronyssinus*; Bt: *B. tropicalis*.

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
