# Peer review of "An Engineered Hybrid Protein from Dermatophagoides pteronyssinus Allergens Shows Hypoallergenicity"

_ijms, 2019, doi:10.3390/ijms20123025_

Round 1

Reviewer 1 Report

A very interesting manuscript

Page 10, Line 262 Authors should specify the name of the company providing the extract used for SPT or if they used their home made extract  

Author Response

Point 1. Page 10, Line 262 Authors should specify the name of the company providing the extract used for SPT or if they used their homemade extract.

Response 1: The following sentence has been added:

For SPT a D. pteronyssinus extract obtained from Greer Laboratories (Lenoir, NC, USA), at 10,000 AU/mL was used. (Page 9, lines 334-333).

Reviewer 2 Report

The topic of the work “An engineered hybrid protein from Dermatophagoides pteronyssinus allergens shows hypoallergenicity” (Manuscript ID: ijms-497221) focuses on a current and interesting subject relative to allergen specific immunotherapy in asthma and rhinitis. Authors show clear results and figures, and their description in the text is correct and easy to follow. However, there are some concerns regarding this paper, so I would recommend the publication of this work, provided that a series of problems are previously addressed.

Major problems

·      Methods require a better and original description.

·      Some sentences are too long and complex. They should be modified a little bit. Here there are a few examples:

o   Lines 48-49: “Der p 1, a cysteine protease and Der p 2, a homolog of the MD-2 adapter molecule, are major 49 allergens inducing sensitization in 50-90% of HDM-allergic individuals [17-19]; therefore, they are……”

o   Lines 166-169: “Our own studies and those from other authors show that allergen combinations (e.g., Der p 1, Der p 2 and a few additional molecules like group 10 mite allergen) increased the diagnostic sensitivity to detect mite allergic patients. This fact guided our molecular design of the hybrid protein [24].”

o   Lines 170-173: “Der p 7 is an allergen generally exhibiting low levels and frequency of IgE binding. However, this allergen has been reported to be structurally related to a protein in the Toll like receptor (TLR) pathway similar to lipid binding protein (LBP), which can interact with dendritic cells activated with LPS.” Sentence too long. Bibliographic citations do not appear.

o   Lines 245-246: “.. to define the source of IL-10 because several cells present 245 in PBMC also produce IL-10 [46]. Therefore, the ability of DPx4 to stimulate a particular IL-10 producing cell remains to be stablished.”

o   Lines 422-423: “DPx4-specific IgG1 and IgG2a in serum were determined by ELISA by means of MaxiSorp plate wells (Nunc,Thermo Fisher), which were coated with 1 μg/mL of DPx4.”

·       Lines 83-85: “The specific IgE levels to the hybrid protein were significantly lower than those to D. pteronyssinus extract (Figure 3A), Der p 1 and Der p 2 (p < 0.001), in 32 samples randomly selected from 90 sera (Figure 3B).” This sentence does not correspond to Figures 3A and 3B. In the figure caption these results appear well explained, but not in the main text. Redo the sentence and differentiate the data presented in Figure 3A and Figure 3B, as well as the statistical significance of each of them.

·       Figure 3: Authors have used values of optical density and not of concentration of IgE (Figure 3A and B). Use of a calibration curve with pure human IgE is advised

·       Figure 5: Specific enzymatic activity values for papain (C+), DPx4, and FABP-4 should be shown in text instead of using a substrate concentration vs OD units plot. If authors chose a plot, reaction rate (Y axis) is preferred. Km of papain for PFLA is about 0.3-0.4 mM, which means that saturating substrate concentrations are not used (authors should use at least 10 x Km to get close to the Vmax). Der p1 is not used as a positive control, which would be the most interesting than papain to make a comparison with DPx4. Lines 389-397: To measure protease activity a kinetic assay using the chromogenic substrate PFLNA was used, but reaction temperature was not indicated.

·       Figure 6: Technique replicates are absent in every patient, with no mean and SD data. Figure caption should be “Basophil activation test. Stimulation index obtained in samples from seven allergic patients stimulated with 10 μg/mL anti-IgE, Dp (extract), Der p2, and DPx4.”

·       Figure 7: These results are clear regarding IL-10 and DPx4. Sample size is n=6, but are they PBMCs from 6 different individuals with allergic asthma or 6 technical replicates from one single donor? Please clarify

·       Table 1: Optic density values? nm?

·       Lines 182-186: “Recently, after the design and obtaining of DPx4 molecule, Der p 23 was described as an allergen inducing high frequencies of sensitization in allergic people from Europe (Austria, France and Germany) [27] and USA [28] and suggested as allergen to be considered for immunotherapy of mite allergy. It seems reasonable that the IgE reactivity displayed by this allergen in the referred populations merits to include Der p 23 in design of novel hybrid molecules.” This sentence should be moved to the last part of the discussion, because it refers to something that could be done in the future. Sometimes is difficult to follow the "storyline" along the discussion.

·       Lines 211-214: “The reactivity of the anti-Der p 1 monoclonal antibody against DPx4 suggests that the hybrid protein keeps at least one of the B-cell epitopes from the native allergen Der p 1, which involves the 213 a.a residues Glu13, Arg17, Gln18 and Asp198 [33, 34]. The presence of shared epitopes is also supported by the 35.1% of inhibition of IgE reactivity to D. pteronyssinus extract by DPx4 (Figure 4D).” However, this does not happen with Der p2 ..... Any explanation? Could it be linked to the number of segments from Der p1 (4) and Der p2 (2) in DPx4?

·       Lines 247-256: Unify the last two paragraphs of the discussion

·       Check lines 305 and 381

·       Line 402: “bottom plate; Costar, Cambridge, U.K.) at 500.000 cells per well.” Please, cell density should be indicated instead of the absolute number of cells per well.

·       Lines 409-413: “Previous to analysis purified DPx4, and mite extract were subjected to affinity chromatography using the ToxinEraser™ Endotoxin Removal Resin Kit (Piscataway, NJ, USA). The concentration of LPS was determined using the ToxinSensor Chromogenic LAL Endotoxin Assay kit (Piscataway, NJ, USA). The content of LPS in the culture supernatant was 0.07 EU/mL for DPx4 and 0.02 EU/mL for mite extract.” This section should be moved upwards, prior to the description on how levels of IL-5, IL- 4, IL-10, and IFNγ were determined and cell cultures were setup.

·       Lines 430-432: “For comparison of groups, the Mann–Whitney U-test was performed using GraphPad Prism software. (San Diego, CA, USA). Data were expressed as mean ±SD P-values less than 0.05 were considered statistically significant”. Sentence should be redone. The statistical analysis seems not totally correct. Authors have more than two groups, so the nonparametric Mann–Whitney U-test is not the most appropriate when one has more than two groups of independent samples. Kruskall-Wallis with a post-hoc test like Dunn´s is perhaps more suited.

Author Response

Major problems

Point 1. Methods require a better and original description.

Response 1: Changes have been applied to the description of methods.

Point 2. Some sentences are too long and complex. They should be modified a little bit. Here there are a few examples:

o   Lines 48-49: “Der p 1, a cysteine protease and Der p 2, a homolog of the MD-2 adapter molecule, are major 49 allergens inducing sensitization in 50-90% of HDM-allergic individuals [17-19]; therefore, they are……”

Response 2: A shorter sentence has been constructed:

“Der p 1 and Der p 2 are major allergens [17-19] therefore, they are ……. (page 2 line 45),

Point 3. Lines 166-169: “Our own studies and those from other authors show that allergen combinations (e.g., Der p 1, Der p 2 and a few additional molecules like group 10 mite allergen) increased the diagnostic sensitivity to detect mite allergic patients. This fact guided our molecular design of the hybrid protein [24].”

Response 3:  The sentence has been modified:

“The design of the hybrid protein was also guided by reports that combinations of Der p 1, Der p 2 and a few additional allergens, including group 10 mite allergen, increased the probability of appropriate diagnosis [18,25],  (Page 7, lines 203-204).

Point 4 Lines 170-173: “Der p 7 is an allergen generally exhibiting low levels and frequency of IgE binding. However, this allergen has been reported to be structurally related to a protein in the Toll-like receptor (TLR) pathway similar to lipid binding protein (LBP), which can interact with dendritic cells activated with LPS.” Sentence too long. Bibliographic citations do not appear.

Response 4: The sentence has been modified and reference included:

“Der p 7 generally exhibits low levels and frequency of IgE binding. However, it has been reported that can interact with LPS activated dendritic cells, and that might stimulate the innate immune response [26]. (Page 7 lines 206-207).

Point 5. Lines 245-246: “.. to define the source of IL-10 because several cells present 245 in PBMC also produce IL-10 [46]. Therefore, the ability of DPx4 to stimulate a particular IL-10 producing cell remains to be stablished.”

Response 5:  Sentence has been modified:

…”…. to define the source of IL-10 because this cytokine is produced by different cells present in PBMC [45]. Therefore, the type of IL-10 producing cell stimulated by DPx4 remains to be established”. (Page 9, lines 308-309).

Point 6. Lines 422-423: “DPx4-specific IgG1 and IgG2a in serum were determined by ELISA by means of MaxiSorp plate wells (Nunc,Thermo Fisher), which were coated with 1 μg/mL of DPx4.”

Response 6: The sentence was modified (page 13, lines 510):

“DPx4-specific IgG1 and IgG2a in serum were determined by ELISA using MaxiSorp plate wells (Nunc,Thermo Fisher) coated with 1 μg/mL of DPx4………”

Point 7. Lines 83-85: “The specific IgE levels to the hybrid protein were significantly lower than those to D. pteronyssinus extract (Figure 3A), Der p 1 and Der p 2 (p < 0.001), in 32 samples randomly selected from 90 sera (Figure 3B).”

This sentence does not correspond to Figures 3A and 3B. In the figure caption these results appear well explained, but not in the main text. Redo the sentence and differentiate the data presented in Figure 3A and Figure 3B, as well as the statistical significance of each of them.

Response 7:  Sentence has been modified:

“Specific IgE levels to DPx4 were significantly lower than those to D. pteronyssinus extract in 90 serum samples from allergic patients, p< 0.001, (Figure 3B). In 32 sera from random-selected allergic patients the IgE levels to DPx4 were also significantly lower than those to Der p 1 and Der p 2, p< 0.001, (Figure 3C)”. (Page 3, lines 98-101).

Point 8. Figure 3: Authors have used values of optical density and not of concentration of IgE (Figure 3A and B). Use of a calibration curve with pure human IgE is advised

Response 8: Done:

IgE levels have been expressed in concentration (ng/mL). The y-axes of figures were modified, and their numbers changed to figure 3B and 3C, respectively. (Page 3).

An additional sentence has been added to the methods (page 11, lines 434-436):

“Quantification of IgE was performed by using the standard curve of human IgE (Merk Millipore, Billerica, MA, USA). IgE diluted in 0.2 M carbonate buffer at 2-fold dilutions (125 ng/ml to 0.1 ng/mL) was used for coating the wells”.

Point 9. Figure 5: Specific enzymatic activity values for papain (C+), DPx4, and FABP-4 should be shown in text instead of using a substrate concentration vs OD units plot. If authors chose a plot, reaction rate (Y axis) is preferred. Km of papain for PFLA is about 0.3-0.4 mM, which means that saturating substrate concentrations are not used (authors should use at least 10 x Km to get close to the Vmax). Der p1 is not used as a positive control, which would be the most interesting than papain to make a comparison with DPx4. Lines 389-397: To measure protease activity a kinetic assay using the chromogenic substrate PFLNA was used, but reaction temperature was not indicated.

Response 9: In attention to this comment, we calculated the protease activity from the initial rate of GPLNA hydrolysis by determining pNA liberation within a given period. The release of the substrate was monitored by spectrophotometry at 410 nm. The typo mistake “420 nm” was corrected.

The initial figure 5 was removed and the results are shown in the text, with this paragraph: “The cysteine protease assay showed the following value of specific enzymatic activity: 67.040 x 10-4 µMol/min for papain, 48.691 µMol/min for DPx4 and 76.5 µMol/min for FABP-4, indicating that DPx4 has around 7,000 fold less activity compared to papain used as positive control. The enzymatic activity of DPx4 was similar to that obtained with the recombinant FABP4, a protein lacking protease activity” (page 4, lines 141-144). 

The temperature used in the assay was 400C, as was indicated in the methods (page 13, line 481).

Papain is a valid control reagent for cysteine protease activity, which we were testing for. Although Der p 1 would be a nice choice in the context of this work, we could not access to the amount of purified Der p 1 necessary for this assay.

Point 10. Figure 6: Technique replicates are absent in every patient, with no mean and SD data. Figure caption should be “Basophil activation test. Stimulation index obtained in samples from seven allergic patients stimulated with 10 μg/mL anti-IgE, Dp (extract), Der p 2, and DPx4.

Response 10:  In this revised version these data are presented in Figure 5, and caption was modified as recommended (page 5).

Point 11. Figure 7: These results are clear regarding IL-10 and DPx4. Sample size is n=6, but are they PBMCs from 6 different individuals with allergic asthma or 6 technical replicates from one single donor? Please clarify.

Response 11: In this revised version these results are presented in Figure 6. Sample size was PBMC from six different individuals with allergic asthma. The figure caption has been modified to clarify (page 5).

Point 12. Table 1: Optic density values? nm?

Response 12: They are O.D. values. This information has been added to table 1. A footnote was also added to clarify (Page 8).

Point 13. Lines 182-186: “Recently, after the design and obtaining of DPx4 molecule, Der p 23 was described as an allergen inducing high frequencies of sensitization in allergic people from Europe (Austria, France and Germany) [27] and USA [28] and suggested as allergen to be considered for immunotherapy of mite allergy. It seems reasonable that the IgE reactivity displayed by this allergen in the referred populations merits to include Der p 23 in design of novel hybrid molecules.” This sentence should be moved to the last part of the discussion because it refers to something that could be done in the future. Sometimes is difficult to follow the "storyline" along the discussion.

Response 13: The mentioned sentence was moved to the last part of the discussion (page 8, lines 320-324).

Point 14. Lines 211-214: “The reactivity of the anti-Der p 1 monoclonal antibody against DPx4 suggests that the hybrid protein keeps at least one of the B-cell epitopes from the native allergen Der p 1, which involves the a.a residues Glu13, Arg17, Gln18 and Asp198 [33, 34]. The presence of shared epitopes is also supported by the 35.1% of inhibition of IgE reactivity to D. pteronyssinus extract by DPx4 (Figure 4D).” However, this does not happen with Der p2 ..... Any explanation? Could it be linked to the number of segments from Der p1 (4) and Der p2 (2) in DPx4?

Response 14: In order to explain, the following sentence has been added:

“Anti Der p 2 monoclonal antibody (Dpx-A9) reacts also against DPx4 (Figure S3), indicating the presence of specific epitopes. The residues 47 to 52 of Der p 2, which belong to an epitope recognized by this monoclonal [33], are present on DPx4. However, the inhibition of IgE reactivity to Der p 2 by DPx4 was 26%, compared to the 38% of inhibition obtained with Der p 1 (Figure 4A and 4B). (Pages 8, lines 260-265). This difference can be explained by a lower percentage of Der p 2 in the structure of DPx4 compared to Der p 1.

The reference 33 was included: Mueller G, et-al. Hydrogen exchange nuclear magnetic resonance spectroscopy mapping of antibody epitopes on the house dust mite allergen Der p 2. The Journal of Biological Chemistry. 2001. 276; 9359-9365.

Point 15. Lines 247-256: Unify the last two paragraphs of the discussion

Response 15: Done

Point 16. Check lines 305 and 381

Response 16: lines 305 and 381 checked and edited for correction.

Point 17. Line 402: “bottom plate; Costar, Cambridge, U.K.) at 500.000 cells per well.” Please, cell density should be indicated instead of the absolute number of cells per well.

Response 17: Absolute number was replaced by cell density:  

“,,,,,at 1x106 /mL per well in 500 µL of RPMI”.  (Page 13, Line 495).

Point 18. Lines 409-413: “Previous to analysis purified DPx4, and mite extract were subjected to affinity chromatography using the ToxinEraser™ Endotoxin Removal Resin Kit (Piscataway, NJ, USA). The concentration of LPS was determined using the ToxinSensor Chromogenic LAL Endotoxin Assay kit (Piscataway, NJ, USA). The content of LPS in the culture supernatant was 0.07 EU/mL for DPx4 and 0.02 EU/mL for mite extract.” This section should be moved upwards, prior to the description on how levels of IL-5, IL- 4, IL-10, and IFNγ were determined and cell cultures were setup.

Response 18: The mentioned sentence has been moved upwards, as recommended (page 13, lines 488-491).

Point 19. Lines 430-432: “For comparison of groups, the Mann–Whitney U-test was performed using GraphPad Prism software. (San Diego, CA, USA). Data were expressed as mean ±SD P-values less than 0.05 were considered statistically significant”. Sentence should be redone. The statistical analysis seems not totally correct. Authors have more than two groups, so the nonparametric Mann–Whitney U-test is not the most appropriate when one has more than two groups of independent samples. Kruskall-Wallis with a post-hoc test like Dunn´s is perhaps more suited.

Response 19: The sentence of statistical analysis was redone:

In this study, we performed paired comparisons on data with a non-normally distributed.  The new paragraph is: “Statistical analyses were performed using Graph Pad Prism software (San Diego, CA. USA). Data were expressed as mean ±SD. P-values less than 0.05 were considered statistically significant. For 2-group comparisons of paired continuous variables, Wilcoxon rank test was used. Friedman’s test was used for multiple repeated measure comparisons” (page 14, lines 530-533).

Reviewer 3 Report

The manuscript by Martínez et al. describes production and characterization of a chimeric molecule that was designed based on the sequences of four allergens originating from Dermatophagoides pteronyssinus. The authors performed the studies to evaluate allergenic properties of the chimeric protein in a search of recombinant proteins that may be used for immunotherapy of house dust mite allergic individuals. The studies are well designed and described. However there are some fragments that require some additional explanations/clarification.

1. Lines 58-59

The authors state that the protein was successfully refolded. However this statement is most likely not true, as the structure of the chimeric molecule is unknown. Most likely it is better to state that the protein was solubilized. A few lines later (lines 64-66) the authors write that the CD spectrum suggests that the protein is only partially folded, which is in clear conflict with the previous statement.

2. The authors should show as a control CD spectra of natural and/or recombinant Der p 1 – it should be added to Figure 1C.

3. A full sequence of the hybrid protein (including purification tag) should be provided in the Supplementary Materials. In addition the authors should be specific which isoallergens were taken into account in the case of Der p 1 and Der p 2. Please use the full nomenclature names for the allergens that are described in the manuscript.

4. Was there any attempt to purify the hybrid protein using size exclusion chromatography?

5. Figure 2 should also include a view of the recombinant protein in a surface representation to show whether the sequence blocks originating from various allergens are forming on the surface patches that indeed may correspond to potential conformational epitopes.

6. Please provide an additional comment on Figure 3 and the fact that a relatively small fraction of the shown sera react with Der p 1.

7. Lines 179-181. Please provide information about which structural parameters were improved or were monitored during the design of the hybrid molecule.

8. Lines 339-340

Please add to the supplementary materials a table that will summarize information on the secondary structure content of the design protein. This table should have information on the secondary structure derived from CD spectra, as well as from the model obtained for the hybrid molecule. In addition, for comparison it should have information on the Der p 1 secondary structure.

9. Figure S3 caption should list explicitly the names of the monoclonal antibodies that were used.

10. Please comment on the size of molecules characterized by DLS. How does it compare with the size of Der p1?

Author Response

Reviewer 3.

Point 1. Lines 58-59

The authors state that the protein was successfully refolded. However, this statement is most likely not true, as the structure of the chimeric molecule is unknown. Most likely it is better to state that the protein was solubilized. A few lines later (lines 64-66) the authors write that the CD spectrum suggests that the protein is only partially folded, which is in clear conflict with the previous statement.

Response 1: We agree with the reviewer and changed the sentence to: “It was successfully solubilized by successive dialysis against L-Cysteine-Cystine buffer with L-Arginine.”

Point 2. The authors should show as a control CD spectrum of natural and/or recombinant Der p 1 – it should be added to Figure 1C.

Response 2: We agree with the reviewer that a control CD of Der p 1 would be useful. We obtained the CD spectrum of nDer p 1 in a previous project and include it for comparison in Figure 1C. Furthermore, we added a sentence in line 62-64 of the original paper to indicate the differences in secondary structure between DPx4 and rDer p 1. This difference is also evidenced by the new Table S1 in the Supplementary Materials (see also answer to point 8).

Point 3. A   sequence of the hybrid protein (including purification tag) should be provided in the Supplementary Materials. In addition the authors should be specific which isoallergens were taken into account in the case of Der p 1 and Der p 2. Please use the full nomenclature names for the allergens that are described in the manuscript.

Response 3: Full amino acid sequence is provided in the supplement (Figure S4).

Full nomenclature name allergens have been added for Der p 1.01, Der p 2.01, Derp 7.01 and Der p 10.01 ( Page 10 lines 481-382)

Point 4. Was there any attempt to purify the hybrid protein using size exclusion chromatography?.

Response 4: Yes, we made purification using size exclusion chromatography following of ion exchange chromatography with Mono Q column (Bio-Rad), with good results. However, this method was used after our initial work using affinity column purification presented in this manuscript. Therefore, there are no data pertinent to this manuscript.  

Point 5. Figure 2 should also include a view of the recombinant protein in a surface representation to show whether the sequence blocks originating from various allergens are forming on the surface patches that indeed may correspond to potential conformational epitopes.

Response 5: Done.  The surface model has been included in figure 2. (Page 2).

Point 6. Please provide an additional comment on Figure 3 and the fact that a relatively small fraction of the shown sera reacts with Der p 1.

Response 6: A comment has been added (page 4, lines 94-96): “The small number of sera with IgE reactivity to Der p 1 is unexpected because a higher frequency of reactivity to this allergen in mite allergic patients from this region has been reported [18]. However, the low number of sera used in the dot blot assay (n = 26) could be one of the reasons for this result”.

Point 7. Lines 179-181. Please provide information about which structural parameters were improved or were monitored during the design of the hybrid molecule.

Response 7: The following sentence has been added: 

 “such as the increase of the number of amino acids in favored regions according to the Ramachandran plot, and better Z-score value for global and local quality prediction of the structure according to ProSA-web [23]” (Page 7, lines 214-216)

Point 8. Lines 339-340

Please add to the supplementary materials a table that will summarize information on the secondary structure content of the design protein. This table should have information on the secondary structure derived from CD spectra, as well as from the model obtained for the hybrid molecule. In addition, for comparison it should have information on the Der p 1 secondary structure.

Response 8: We have added Table S1 to the supplementary materials containing the secondary structure content calculated from the CD spectrum of DPx4, the 3D structure of Der p 1 (PDB 3F5V) and the homology model of DPx4.

Point 9. Figure S3 caption should list explicitly the names of the monoclonal antibodies that were used.

Response 9:   The names of the monoclonal antibodies have been listed in the figure S3 caption.

Point 10. Please comment on the size of molecules characterized by DLS. How does it compare with the size of Der p1?

Response 10: In order to clarify the estimated degree of aggregation of DPx4 the following sentence was added in the results section:

“From DLS data it can be estimated that aggregates consist of approximately 700 to 2000 molecules of DPx4 in the reduced and the oxidized state, respectively. In contrast, the natively folded Der p 1 has been shown to exist as a monomer in solution [22]. (Page 2 Lines 58-60)

Round 2

Reviewer 2 Report

Regarding the manuscript “An engineered hybrid protein from Dermatophagoides pteronyssinus allergens shows hypoallergenicity” (Manuscript ID: ijms-497221), authors have corrected the major problems of the work. They have answered all the questions previously asked by this reviewer as well. Therefore, I recommend to editors the publication of this work in the current state.